# The AID study: protocol for a randomised controlled trial of intrauterine insemination in the natural cycle compared with intracervical insemination in the natural cycle

Petronella Kop,[1] Madelon van Wely,[1] Annemiek Nap,[2] Ben Willem Mol,[3] Rob Bernardus,[4] Michael De Brucker,[5] Pim Janssens,[6] Ben Cohlen,[7] Jacqueline Pieters,[8] Sjoerd Repping,[1] Fulco van der Veen,[1] Monique H Mochtar[1]

For numbered affiliations see end of article.

**Correspondence to**
Mrs Petronella Kop;
p.a.kop@amc.uva.nl

## ABSTRACT

**Introduction** At present, studies comparing intrauterine insemination in the natural cycle versus intracervical insemination in the natural cycle in women undergoing artificial insemination with donor sperm are scarce.

**Methods and analysis** We perform a randomised controlled non-inferiority trial among five secondary and tertiary fertility clinics in the Netherlands and one tertiary fertility clinic in Belgium. Women eligible for artificial insemination with donor sperm are included. We perform six cycles of artificial insemination with donor sperm within a time horizon of 8 months comparing intrauterine insemination in the natural cycle with intracervical insemination in the natural cycle. The primary outcome is ongoing pregnancy leading to live birth conceived within eight months after randomisation. Secondary outcomes are clinical pregnancy rate, miscarriage rate, multiple pregnancy rate, pregnancy complications (preterm birth, birth weight <2500 g, pregnancy induced hypertension, (pre-) eclampsia, Hemolysis Elevated Liver enzymes Low Platelets (HELLP)), time to ongoing pregnancy, direct and indirect costs. To demonstrate the non-inferiority of intracervical insemination with a margin of 12%, we need 208 women per arm.

**Ethics and dissemination** The study has been approved by the Medical Ethical Committee of the Academic Medical Centre and from the Dutch Central Committee on research involving human subjects (47330-018-13). The boards of the participating hospitals approved the study. Results will be disseminated through peer-reviewed publications and presentations at international scientific meetings.

**Trial registration number** NTR4462

## INTRODUCTION

Artificial insemination with donor sperm (AID) may be performed in heterosexual couples for medical reasons or to assist lesbian couples or single women to achieve pregnancy. Medical reasons for AID include obstructive and non-obstructive azoospermia, severely impaired semen quality in couples who do not

### Strengths and limitations of this study

► A multicentre randomised controlled design is used.
► We aim to give cumulative ongoing pregnancy rates, while previous studies reported clinical pregnancy rates.
► A limitation is that participants and researchers were not blinded.

wish to undergo or were not successful with intracytoplasmatic sperm injection (ICSI), severe rhesus iso-immunisation, prevention of vertical transmission of a genetic defect or prevention of transmission of HIV in couples who do not wish to undergo semen washing.[1 2] Women can be inseminated via the intrauterine (IUI) or the intracervical (ICI) route with or without ovarian stimulation.

A systematic review, in women who started AID, reported IUI with ovarian stimulation (IUI-OS) to be more effective than ICI-OS in terms of live birth rate (OR 2.55, 95% 0.72 to 8.96, 1 RCT, n=43) and clinical pregnancy rate (OR 2.83, 95% CI 1.38 to 5.78, 2 RCTs, n=131) at the cost of a higher multiple pregnancy rate (OR 2.77, 95% CI 1.00 to 7.69, 2 RCTs, n=131). In women who started AID in the natural cycle, IUI resulted in similar live birth rates as ICI in the natural cycle (OR 3.24, 95% 0.12–87.13, 1 RCT, n=26) but in higher clinical pregnancy rates in (OR 6.18, 95% CI 1.91 to 20.03, 2 RCTs, n=76). There were no multiple pregnancies.[3]

Multiple pregnancies are associated with increased perinatal and maternal morbidity and perinatal mortality and are regarded as an adverse treatment outcome that is no longer acceptable and should be prevented.[4–6]

Therefore, the single existing guideline on AID, published by The Royal College of Obstetrics and Gynaecology, recommends six cycles of IUI in the natural cycle.[1]

In 2012, we performed a retrospective nationwide cohort study among eight sperm banks in the Netherlands on IUI and ICI in the natural cycle. We included 1843 women of whom 1163 underwent 4269 cycles of IUI in the natural cycle and 680 underwent 2345 cycles of ICI in the natural cycle. Cumulative ongoing pregnancy rates over six treatment cycles were 40.5% for IUI and 37.9% for ICI, resulting in an HR of 1.02 (95% CI 0.84 to 1.23) for IUI versus ICI.[7]

The costs for IUI have been estimated to be four times higher than ICI, mostly because of the sperm processing inherent to IUI.[8]

Since evidence that underpins the superiority of IUI in the natural cycle over ICI in the natural cycle is scarce and IUI may generate higher costs than ICI for no increase in pregnancies, we propose a multicentre trial comparing IUI and ICI in the natural cycle with a cost-effectiveness analysis alongside the trial.

## OBJECTIVE

To compare the effectiveness of six cycles of IUI in the natural cycle compared with six cycles of ICI in the natural cycle with cryopreserved donor sperm within a time horizon of eight months.

## METHODS AND ANALYSIS
### Study design

This study is a non-blinded, multicentre, non-inferiority randomised controlled trial among five secondary and tertiary fertility clinics in the Netherlands and one tertiary fertility clinic in Belgium. Recruitment started on 3 June 2014.

We expect to complete recruitment on 1 January 2020.

### Study population
#### Inclusion criteria

We will study women between 18 and 43 years with a regular cycle, be it spontaneously or after ovulation induction and with an indication for AID.

Indications for AID are medical reasons like obstructive and non-obstructive azoospermia, severely impaired semen quality in couples who do not wish to undergo or were not successful with ICSI, severe rhesus iso-immunisation, prevention of vertical transmission of a genetic defect or prevention of transmission of HIV in couples who do not wish to undergo semen washing, In addition, lesbian couples or single women who apply for AID are eligible to participate.

#### Exclusion criteria

Women with known double-sided tubal pathology, irregular menstrual cycles, in-vitro fertilisation or IUI in their history will not be eligible.

### Randomisation

Randomisation is performed by accessing a web-based data system that is used for randomisation in clinical trials and will be performed centrally with the use of a permuted-block-design with randomly selected block sizes that vary between two, four and six. We used a permuted block design to randomly allocate participants to a treatment and aimed to maintain balance across treatment groups. In this way, the clinician who performs the randomisation cannot predict the next randomisation to follow. Patients are randomised individually. Patients are randomly allocated to either six cycles of IUI in the natural cycle or six cycles of ICI in the natural cycle.

### Interventions

In the IUI cycles, ovulation detection is performed with Luteinising Hormone (LH) tests or by transvaginal sonography, depending on the local setting. In case of monitoring with urinary LH tests, women test their urine once per day, starting on an individually calculated cycle day based on their basal body temperature chart. Women are inseminated with processed semen the day after the endogenous LH surge has been detected in the urine sample. Serum LH measurements are performed from day 11 of the menstrual cycle onwards and women are inseminated 1 day after the serum LH rise.[9] In case follicular growth is monitored by transvaginal sonography, timing of insemination can be performed by ovulation triggering or by measuring serum LH levels. In case of ovulation triggering human chorionic gonadotropin (Pregnyl, Organon, Oss, The Netherlands) is administered when the dominant follicle has a diameter of at least 16 mm. Women are inseminated 36 to 40 hours thereafter. The semen may be processed in two ways. One way is to process the semen against a density gradient centrifugation and/or a washing step with culture medium according to local laboratory protocol after thawing of the—unprocessed—cryopreserved semen and the second way is to process the semen against a density gradient centrifugation and/or washing step with culture medium according to local laboratory protocol before freezing of the semen. In the ICI cycles, ovulation detection is in the same way as in the IUI arm and insemination is also timed identical to the IUI arm. Women are inseminated with unprocessed semen once per cycle, and preferably by a cervical cap.[6] Before starting a new treatment cycle with IUI or ICI, women undergo a pregnancy test (Human gonadotrophin hormone (hCG) measurement in serum or urine). If the woman is not pregnant the next treatment cycle is started according to protocol. In case of a positive pregnancy test, women are monitored by transvaginal sonography. Monitoring will take place at 5 to 9 weeks of amenorrhoea to check whether an intrauterine gestational sac is present, that is, a clinical pregnancy.

Subsequently, monitoring takes place at 11 to 12 weeks amenorrhoea to register the presence of an intrauterine gestational sac with fetal heartbeat, that is, an ongoing pregnancy.

## Follow up

Pregnant women undergo an ultrasound at 7 and 11 weeks of gestation to classify the pregnancy as clinical or ongoing singleton or multiple pregnancy. Women are contacted by telephone to enquire on the course of pregnancy, delivery and the health of the child. Detailed information on maternal complications will be obtained from the responsible obstetrician or midwife. When data on the health of the child are insufficient, child health centres and/or paediatricians are contacted for specific information. Presumably, not all women complete the 8 months of treatment. Couples who drop-out will largely represent normal patient flow. In both arms, women will be treated until ongoing pregnancy occurs within a time horizon of 8 months. We aim to keep track of all couples who drop-out and to document the reason for the drop-out.

## Patient and public involvement

The Dutch patient organisation for patients with fertility problems (Freya) supports the study. When the study is finished, we will request Freya to organise an informal meeting during which fertility experts from our study group will present information on the treatment options. We will ask Freya to publish an easy to read summary of the results on their homepage. The data will thus become available for the public. Study participants are asked if they want to be informed about the results of the study during the informed consent procedure.

## Primary outcome measure

The primary outcome is ongoing pregnancy leading to live birth. Ongoing pregnancy is defined as a positive heartbeat at 12 weeks of gestation. Only ongoing pregnancies that occur within the first eight months after randomisation count for assessment of the primary endpoint.

## Secondary outcome measures

Secondary outcomes are clinical pregnancy (any registered heartbeat at ultrasound), miscarriage (registered heartbeat before 12 weeks of gestation) and multiple pregnancy (registered heartbeat of at least two fetuses at 12 weeks of gestation) and ectopic pregnancy, time to ongoing pregnancy leading to live birth.

We will register the following pregnancy outcomes: preterm birth <37 weeks, iatrogenic preterm birth, spontaneous preterm birth, time to delivery, mean birth weight, birth weight <2.5 g, growth restriction defined as birth weight <10th percentile, pregnancy-induced hypertension, pre-eclampsia, HELLP syndrome, and direct and indirect costs.

## Power calculation

The study is designed as a non-inferiority trial. We assume that six cycles can be completed within a fixed time horizon of 8 months. Assuming a live birth rate of 40% after six cycles of IUI, we need 208 women per arm with an alpha of 0.05, and a beta of 0.80 to demonstrate the non-inferiority of ICI with a margin of 12%.[7]

The margin of 12% are the differences shown in literature between IUI and ICI.[3]

A limitation of our study is that we use a non-inferiority margin of 12%. Most clinicians and patients would feel that 12% is a rather large and clinically relevant difference.

## Data analysis

We will compare baseline measurements female age, body mass index, indication for AID, previous assisted reproductive treatments, previous pregnancies, parity, intoxication and referral status.

The analysis of all outcomes is on an intention-to-treat basis. For ongoing pregnancy we will test non-inferiority on basis of the absolute risk difference with the absolute margin of 12%. We will subsequently estimate differences in the primary and secondary outcomes as relative risks with 95% CI. We will also report p values.

We will evaluate association and interaction with our primary outcome of baseline variables in logistic regression analyses. The baseline variables that we plan to include in these analyses are timing, sperm quality and practice variation.

We will construct Kaplan-Meier survival curves for time to ongoing pregnancy.

For continuous variables, we examine the distribution of the observations, and if normally distributed, we summarise them as means with SDs. If they are not normally distributed, medians and IQRs are reported. For dichotomous data, we provide proportions (or percentages). In addition to the baseline and outcome data, we also summarise the recruitment numbers, those lost to follow-up, protocol violations and other relevant data. We will analyse a maximum of six cycles of IUI or ICI performed within a time horizon of eight months after randomisation.

## Data safety monitoring board

A data safety monitoring board (DSMB) is installed for this trial. In the DSMB, professionals from the fields of fertility and epidemiology are represented. All members have independent positions from the trial study group. An interim-analysis is performed on the primary endpoint when 200 patients have been randomised and all have completed at least 3 months of follow-up to enable retrieval of the outcome ongoing pregnancy. The interim-analysis will be performed by an independent statistician, blinded for the treatment allocation. The statistician will report to the independent DSMB. The DSMB will decide on the continuation of the trial. There are no safety issues liaised to this trial and futility is welcomed as an outcome. The only reason for prematurely stopping the trial is a large difference in ongoing pregnancies. We suggest the Peto approach to be used: if an interim analysis shows a probability of less than 0.001 that the treatments are different in terms of ongoing pregnancy, then the trial should be stopped early.

## Economic evaluation

We will perform an economic analysis from a health-care perspective alongside the clinical trial. We make a distinction between direct costs and indirect costs. Direct costs include the costs of medication, cycle monitoring, interventions and the costs of pregnancy leading to live birth. Indirect costs are collected from the individual case report forms of the RCT. For each woman, we registered cycle monitoring (number of ultrasounds), we registered the medication and interventions (cycles with IUI, cycles with ICI) they received within six subsequent cycles or until an ongoing pregnancy occurred within a time horizon of 8 months.

Societal costs of travel and time will be determined on basis of recourse use as registered in the case report form.

We will analyse a cost-minimisation or cost-effectiveness analysis depending on the outcome of ongoing pregnancy rates in both groups. We present the cost-effectiveness of each strategy as cost per ongoing pregnancy and costs per live birth. We explore the robustness of the results for various assumptions and parameter estimates in sensitivity analysis outcomes and we express these in incremental cost-effectiveness ratio graphs and cost-effectiveness acceptability curves. The economic evaluation will be reported in a separate paper.

## DISCUSSION

In women who start AID, there is uncertainty about the most cost-effective treatment modality. IUI-OS leads to higher clinical pregnancy rates compared with ICI-OS, but the incidence of multiple pregnancies with the addition of OS is substantial and hardly acceptable.[3] Studies on IUI and ICI in the natural cycle report no differences in live birth rate between IUI and ICI in natural cycles, but higher clinical pregnancy rates for IUI, without any multiple pregnancies. Therefore, the National Institute for Health and Care Excellence guideline recommends IUI in the natural cycle for the first six cycles.[1] Since IUI compared with ICI in natural cycles shows no difference in live birth rate, IUI may generate higher costs than ICI for no increase in live birth rates. There are no large randomised controlled trials that addresses this issue by comparing IUI in the natural cycle with ICI in the natural cycle in women who start with AID.

### Ethics and dissemination

Results will be disseminated through peer-reviewed publications and presentations at international scientific meeting.

**Author affiliations**
¹Center for Reproductive Medicine, Amsterdam Reproduction & Development Institute, Amsterdam UMC, University of Amsterdam, Amsterdam, Netherlands
²Department of Gynaecology and Obstetrics, Rijnstate, Arnhem, , Netherlands
³Department of Obstetrics and gynaecology, Monash University Central Clinical School, Melbourne, Victoria, Australia
⁴Fertility clinic, Nij Barrahus, Wolvega, Netherlands
⁵Centre for Reproductive Medicine, Universitair Ziekenhuis Brussel, Brussel, Belgium
⁶Clinical Chemistry and Haematology, Hospital Rijnstate, Arnhem, , Netherlands
⁷Obstetrics and Gynaecology, Isala Hospitals, Zwolle, , Netherlands
⁸Fertility clinic, Vivaneo Medisch Centrum Kinderwens, Leiderdorp, Netherlands

**Contributors** PALK is responsible for the overall logistical aspects of the trial and drafted the paper. MvW, MHM, FvdV, SR and BWM designed the trial and were responsible for the development of the protocol. PALK, REB, MB, PMWJ, AN, BJC, JJPMP, MvW, MHM, SR, BMW and FvdV contributed to the protocol included patients and approved the final version of the paper.

**Funding** This trial will receive funding from the Netherlands Organization for Health Research and Development (ZonMw project number 837002407)). There is no role in the study for ZonMw.

**Competing interests** No, there are no competing interests for any author.

**Patient consent for publication** Not required.

**Ethics approval** The study has been approved by the Medical Ethical Committee of the Academic Medical Centre and from the Dutch Central Committee on research involving human subjects (47330-018-13). The boards of the participating hospitals approved the study.

**Provenance and peer review** Not commissioned; externally peer reviewed.

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
