## [Reviewer comments · BMJ Open]

ARTICLE DETAILS

TITLE (PROVISIONAL)	The AID study: protocol for a randomised controlled trial of intrauterine insemination in the natural cycle compared to intracervical insemination in the natural cycle.
AUTHORS	Kop, Petronella; van Wely, Madelon; Nap, Annemiek; Willem Mol, Ben; Bernardus, R.E.; Brucker, M.; Janssens, Pim; Cohlen, B.J.; Pieters, Jacqueline; Repping, S.; van der Veen, F.; Mochtar, Monique

VERSION 1 - REVIEW

REVIEWER	Michael h. Dahan McGill University, Canada
REVIEW RETURNED	04-Sep-2018

GENERAL COMMENTS	This is a research proposal of a non inferiority trial for ICI and IUI in natural cycles. An important study. I have a few issues I would like addressed 1) Please explain why a permuted-block-design with randomly selected block sizes that vary between two, four and six was used as opposed to just randomizing each patient individually. 2) The semen may be processed in two ways. One way is to process the semen against a density gradient centrifugation and/or a washing step with culture medium according to local laboratory protocol after thawing of the –unprocessed- cryopreserved semen and the second way is to process the semen against a density gradient centrifugation and/or washing step with culture medium according to local laboratory protocol before freezing of the semen. Please make a plan to explain how you will deal with the two different types of sperm preparation. 3) You state that ICI may be cheaper than IUI since sperm preparation is different. However, in the study you prepare the sperm similarly for IUI and ICI. How will you know that if the sperm was prepared differently for ICI that results of the comparison would be the same? Please justify in the proposal. 4) Please provide a plan for interim evaluation of the data, and when to stop the trial if results are different.
--

REVIEWER	Christine Elisabeth Skala Klinik für Geburtshilfe und Frauengesundheit der Universitätsmedizin der Johannes Gutenberg Univeristät Mainz, Germany
REVIEW RETURNED	25-Sep-2018

GENERAL COMMENTS	This is a well prepared study protocol, to a relevant clinical question. I am looking forward to seeing the results in a published paper. There is one thing i am missing. There are no data about the donor sperm. Is a good quality of the donor sperm assured?
--

REVIEWER	Emily Liu Fertility PLUS, Auckland District Health Board, New Zealand
REVIEW RETURNED	07-Dec-2018

GENERAL COMMENTS	The proposed study will address the gap in current evidence. The study design and method are appropriate to answer the research question.
---

REVIEWER	Katharine Correia Amherst College, USA
REVIEW RETURNED	20-Feb-2019

GENERAL COMMENTS	This study protocol for a randomized multi-center trial is well laid out and the study is well-designed to address important questions comparing IUI and ICI in natural cycles. I had a few questions/comments:  1. My main concern is about the non-inferiority margin, which is set at 12%. This value seems high to me. In a non-inferiority trial, the non-inferiority margin should not be based on differences observed in the literature, but rather on clinical reasoning (e.g. here, we ask “would a difference of 12% be clinically important?” I would think so.) I realize that decreasing your non-inferiority margin would mean you’d need a larger sample size to declare non-inferiority at the same alpha level, but even a difference of 10% pregnancy rate seems important. If I were a patient, and I knew that if I used one procedure I’d have a 40% chance of pregnancy, and if I used a second procedure, I’d only have a 30% chance of pregnancy, then that would seem a strong enough difference to me that I’d want to choose the former (i.e. I’d consider the latter inferior in its success rate!). It seems like the non-inferiority margin, i.e. the largest clinically acceptable difference, shouldn’t be set any higher than 10%. 2. Can you please clarify the randomization procedure? Will patients be randomized within clinic (blocking by clinic)? (Do the different clinics have similar pregnancy and live birth rates following IUI? Are the pregnancy and live birth rates following ICI similar across the clinics?)
---

	3. Could you provide more specific definitions of direct and indirect costs?
--	--

VERSION 1 – AUTHOR RESPONSE

Comments from Reviewer 1: Michael H. Dahan.

Comment 1.

Please explain why a permuted-block-design with randomly selected block sizes that vary between two, four and six was used as opposed to just randomizing each patient individually.

Answer

We used a permuted block design to randomly allocate participants to a treatment and aimed to maintain balance across treatment groups. In this way, the clinician who performs the randomisation cannot predict the next randomisation to follow.

Comment 2

The semen may be processed in two ways. One way is to process the semen against a density gradient centrifugation and/or a washing step with culture medium according to local laboratory protocol after thawing of the –unprocessed- cryopreserved semen and the second way is to process the semen against a density gradient centrifugation and/or washing step with culture medium according to local laboratory protocol before freezing of the semen.

Please make a plan to explain how you will deal with the two different types of sperm preparation.

Answer

Between the clinics that participate in our RCT there is practice variation between two methods of processing the sperm for IUI. Therefore we will add a sub analysis for practice variation.

We added lines 218-220.

Comment 3

You state that ICI may be cheaper than IUI since sperm preparation is different. However, in the study you prepare the sperm similarly for IUI and ICI. How will you know that if the sperm was prepared differently for ICI that results of the comparison would be the same? Please justify in the proposal.

Answer

Semen for IUI is processed and semen used for ICI is unprocessed.

Line 135-152 describes timing and processing of sperm for IUI.

Line 153- 155 describes timing of ICI and that women are inseminated with unprocessed sperm.

Comment 4

Please provide a plan for interim evaluation of the data, and when to stop the trial if results are different.

Answer

A data safety monitoring board (DSMB) is installed for this trial. In the DSMB professionals from the fields of fertility and epidemiology are represented. All members have independent positions from the trial study group. An interim-analysis is performed on the primary endpoint when 200 patients have been randomised and all have completed at least 3 months of follow-up to enable retrieval of the outcome ongoing pregnancy. The interim-analysis will be performed by an independent statistician, blinded for the treatment allocation. The statistician will report to the independent DSMB. The DSMB will decide on the continuation of the trials. There are no safety issues liaised to this trial and fertility is welcomed as an outcome. The only reason for premature stopping the trial is a large difference in ongoing pregnancies. We suggest the Peto approach to be used: if an interim analysis shows a probability of less than 0.001 that the treatments are different in terms of ongoing pregnancy, then the trial should be stopped early.

We added lines 231-243.

Comments from reviewer 2: Christine Elisabeth Skala

Comment 1

This is a well prepared study protocol, to a relevant clinical question. I am looking forward to seeing the results in a published paper.

There is one thing i am missing. There are no data about the donor sperm. Is a good quality of the donor sperm assured?

Answer

We recognize that the quality of the donor sperm is not mentioned in our protocol.

Guidelines and studies on what good quality of donor sperm is are lacking.

We will register the quality of the donor sperm used for IUI and ICI. We have the intention to perform logistic regression for factors that can influence ongoing pregnancy rates like timing, sperm quality and practice variation (see also comment 2 of reviewer 1 for further sub analysis).

We added lines 218-220.

Comments from reviewer 3: Emily Liu

The proposed study will address the gap in current evidence. The study design and method are appropriate to answer the research question.

Comments from reviewer 4: Katharine Correia

This study protocol for a randomised multi-center trial is well laid out and the study is well-designed to address important questions comparing IUI and ICI in natural cycles. I had a few questions/comments:

Comment 1.

My main concern is about the non-inferiority margin, which is set at 12%. This value seems high to me. In a non-inferiority trial, the non-inferiority margin should not be based on differences observed in the literature, but rather on clinical reasoning (e.g. here, we ask "would a difference of 12% be clinically important?" I would think so.) I realize that decreasing your non-inferiority margin would mean you'd need a larger sample size to declare non-inferiority at the same alpha level, but even a difference of 10% pregnancy rate seems important. If I were a patient, and I knew that if I used one procedure I'd have a 40% chance of pregnancy, and if I used a second procedure, I'd only have a 30% chance of pregnancy, then that would seem a strong enough difference to me that I'd want to choose the former (i.e. I'd consider the latter inferior in its success rate!). It seems like the non-inferiority margin, i.e. the largest clinically acceptable difference, shouldn't be set any higher than 10%.

Answer

A smaller non-inferiority margin is always better, however this is not feasible in our case. Our study started in June 2014 and is still not finished. If we lower the non-inferiority margin, we have to increase the sample size. Increasing the sample size will prolong the study duration and we do not have the financial means to do so.

Comment 2

Can you please clarify the randomization procedure? Will patients be randomized within clinic (blocking by clinic)? (Do the different clinics have similar pregnancy and live birth rates following IUI? Are the pregnancy and live birth rates following ICI similar across the clinics?)

Answer

Patients are randomised individually. We performed a permuted block design to randomly allocate participants to a treatment, while maintaining balance across treatment groups. In this way, the researcher that performs the randomisation is unaware of the next randomisation to come.

We do not stratify by center. This is not required in trials with samples sizes above 200. We are doing a pragmatic trial and our primary effect measure is based on the relative difference between the groups. The relative difference is less likely to be influenced by variations in practise. . We will evaluate association and interaction with our primary outcome of baseline variables in logistic regression analyses. The baseline variables that we plan to include in these analyses are timing, sperm quality and practice variation.

We added this to the protocol in lines 218-220.

Comment 3

Could you provide more specific definitions of direct and indirect costs?

Answer

To provide more specific definitions of direct and indirect costs we changed the paragraph on economic evaluation as follows:

We will perform an economic analysis from a healthcare perspective alongside the clinical trial. We make a distinction between direct costs and indirect costs. Direct costs include the costs of medication, cycle monitoring, interventions and the costs of pregnancy leading to live birth. Indirect costs are collected from the individual case report forms of the RCT. For each woman, we registered cycle monitoring (number of ultrasounds), we registered the medication, and interventions (cycles with IUI, cycles with ICI) they received within six subsequent cycles or until an ongoing pregnancy occurred within a time horizon of eight months. Societal costs of travel and time will be determined on basis of recourse use as registered in the case report form.

We changed lines 247-256.